# Molecular basis for RNA polymerase-dependent transcription complex recycling by the helicase-like motor protein HelD

Timothy P. Newing [1], Aaron J. Oakley [1], Michael Miller [2], Catherine J. Dawson [2], Simon H. J. Brown [1], James C. Bouwer [1], Gökhan Tolun [1✉] & Peter J. Lewis [2✉]

In bacteria, transcription complexes stalled on DNA represent a major source of roadblocks for the DNA replication machinery that must be removed in order to prevent damaging collisions. Gram-positive bacteria contain a transcription factor HelD that is able to remove and recycle stalled complexes, but it was not known how it performed this function. Here, using single particle cryo-electron microscopy, we have determined the structures of *Bacillus subtilis* RNA polymerase (RNAP) elongation and HelD complexes, enabling analysis of the conformational changes that occur in RNAP driven by HelD interaction. HelD has a 2-armed structure which penetrates deep into the primary and secondary channels of RNA polymerase. One arm removes nucleic acids from the active site, and the other induces a large conformational change in the primary channel leading to removal and recycling of the stalled polymerase, representing a novel mechanism for recycling transcription complexes in bacteria.

[1] Molecular Horizons and School of Chemistry and Molecular Bioscience, University of Wollongong, and Illawarra Health and Medical Research Institute, Wollongong, NSW 2522, Australia. [2] School of Environmental and Life Sciences, University of Newcastle, Callaghan, NSW 2308, Australia. ✉email: gokhan_tolun@uow.edu.au; Peter.Lewis@newcastle.edu.au

I n bacteria, transcription and DNA replication occur concomitantly, making potentially damaging collisions of DNA replication forks with transcription complexes inevitable[1–5]. Transcription is highly sensitive to DNA damage, which causes the elongation complex (EC) to pause, and multiple redundant systems have evolved to ensure rapid removal of RNAP and/or the repair of damaged DNA[6–10]. This reduces the chance of replication forks colliding with stalled transcription complexes whilst also serving as an efficient system for maintaining genome integrity, especially within coding regions. However, independent of DNA damage, ~15% of paused transcription complexes are inactive for a significant period of time[11] and require removal through the action of factors such as the transcription recycling factor HelD[12].

HelD is widely distributed in Gram-positive bacteria and has superficial similarity to superfamily 1 (SF1) DNA helicases such as UvrD/PcrA, and its catalytic activity is ATP-dependent[12]. Low-resolution small-angle X-ray scattering data indicate that HelD undergoes an ATP-dependent conformational change and is capable of binding to DNA[13], suggesting that these are important properties of HelD in transcription complex recycling. SF1 helicases UvrD/PcrA bind on the upstream side of RNAP, and are able to reverse-translocate it away from a site of DNA damage[6,14]. Similarly, Rad26 (eukaryotic) and RapA (prokaryotic), Swi2/Snf2 family helicases also bind on the upstream side of RNAP and reverse-translocate stalled complexes during their reactivation[15,16]. Although HelD has been shown to bind on the downstream side of RNAP[12], it seemed reasonable to assume that it may facilitate transcription complex recycling in a similar manner to UvrD/PcrA, RapA and Rad26, utilising ATP-dependent translocation of stalled complexes along a DNA template.

In this work, we show this is not the case and that the structure of HelD enables an unusual mode of transcription complex recycling involving a large conformational change in RNA polymerase (RNAP). Using single particle cryo-electron microscopy (cryo-EM), we determined the structures of Bacillus subtilis RNAP elongation and HelD complexes, enabling analysis of the conformational changes that occur in RNAP driven by HelD interaction. HelD represents a class of motor protein distantly related to the SF1 helicases, containing two arms that flank the helicase-like domains. One arm anchors HelD to RNAP, binding deep within the secondary channel of RNAP where it sterically clashes with nucleic acids in the active site and in doing so distorts the highly conserved bridge helix of RNAP. The other arm pushes open the primary DNA-binding channel of RNAP, causing a conformational change that releases bound DNA. Thus, HelD is a prototypical member of a widely dispersed and divergent branch of the SF1 helicase family that maintain genome integrity by removing non-productive transcription roadblocks.

## Results

### Structure determination of the transcription elongation complex. 
Bacillus subtilis is the model representative organism of the medically and industrially important Firmicutes phylum that have genomes with a low G + C content. Despite considerable effort, no structure of RNAP from the low G + C Gram positives has been determined to date. RNAP core ($\alpha_2\beta\beta'\omega$) was purified (Supplementary Fig. 1), and it's activity established using a HelD-dependent stimulation of multi-round transcription assay which gave identical results to those observed in previous studies[12] (Fig. 1a). We then used cryo-EM to determine the structure of the B. subtilis RNAP transcription elongation complex (EC) at 3.36 Å, to enable an understanding of the conformational changes caused by HelD during transcription complex recycling (Fig. 1b, c, Table 1; Supplementary Fig. 2, Movie 1). RNAP in the Firmicutes

is the smallest multi-subunit polymerase[17,18], and given the industrial and clinical importance of this group of bacteria this complex will serve as an invaluable reference structure.

Despite nuclease treatment of the cell lysate, upon 3D reconstruction of the core structure, nucleic acid was clearly visible indicating that throughout the purification process the core enzyme remained tightly bound to nucleic acids which protected them from nuclease treatment (Supplementary Table 1). Therefore, the structure presented represents an elongation complex (EC) with non-specified nucleic acid sequence (i.e., the reconstructed density shows well-defined ribose-phosphate groups with an average of random base sequences). Typically, the subunit composition of core bacterial RNAP is represented as $\alpha_2\beta\beta'\omega$, but in previous work we identified an additional small subunit called ε was present in B. subtilis RNAP in addition to ω[19]. However, holoenzyme preparations from the strain (LK637, Δ δ; Methods) used in this study lacked ε (Supplementary Fig. 1a) and so the core structure is presented lacking this subunit (although ε is present in the structure of RNAP core in complex with HelD, see below). Previous studies have also shown deletion of ε causes no detectable phenotype or change in gene expression profiles, and RNAP core preparations lacking ε have indistinguishable activity compared to those that do contain it[19,20]. Comparison of EC and RNAP-HelD complexes showed no significant structural differences in the region where ε binds and so in Fig. 1b the ε binding site is indicated as a dotted circle, and in Supplementary Fig. 4a ε is shown as it is clear that RNAP isolated from B. subtilis is a heterogeneous mixture of core ($\alpha_2\beta\beta'\omega$) ± δ, ε, and HelD, in addition to multiple different σ factors[21].

The EC is 150 Å × 112 Å × 123 Å (L × W × H, Table 1), and is broadly comparable to the dimensions of core/elongation complexes from other species (157 × 153 × 136 Å; E. coli 6ALF, 183 × 107 × 115 Å; Mycobacterium smegmatis 6F6W, and 170.1 × 110.1 × 127.8 Å; Thermus thermophilus 2O5I)[22–24], although it appears to be more slender and elongated than the roughly globular E. coli, and shorter than the M. smegmatis and T. thermophilus enzymes due to the lack of insertion sequences (Supplementary Fig. 4).

Due to the high level of sequence conservation amongst RNAPs, the overall structure of the EC was similar to those from other organisms and largely consistent with homology models used in previous work on structure/function studies with B. subtilis RNAP[25–28]. However, modelling had been unable to establish the structure of the ~180 amino acid βln5 insertion in the β2 lobe. The β2 lobe is one of the least well-conserved regions of bacterial RNAPs, and is a hot-spot for the presence of lineage-specific insertions[17] (Supplementary Fig. 4a). The only other region that was significantly different to other bacterial RNAPs was a 10 amino acid loop from β E696-G705 that protrudes from the bottom of the enzyme (Fig. 1c, Supplementary Fig. 4a). Refinement and building sequence into the resulting density indicated the B. subtilis β2 lobe is a continuous globular structure and that the βln5 insertion increases the size asymmetry between the β lobes compared to other Gram positive RNAPs such as those from M. smegmatis and M. tuberculosis[23,29] (Fig. 1b, Supplementary Fig. 4). Searches using DALI[30] found no structural matches to the βln5 insertion leaving its function similarly enigmatic to those of most other lineage-specific insertions.

The absence of lineage-specific insertions perhaps helps to account for the additional subunits found in the Firmicutes such as δ and ε, and this and the accompanying paper by Pei et al.[31] identify the location of these subunits. This suggestion is potentially supported by examination of the T. thermophilus structure around its βln10 and βln12 insertions that localise to a region very close to the ε binding site. Superimposition of ε into

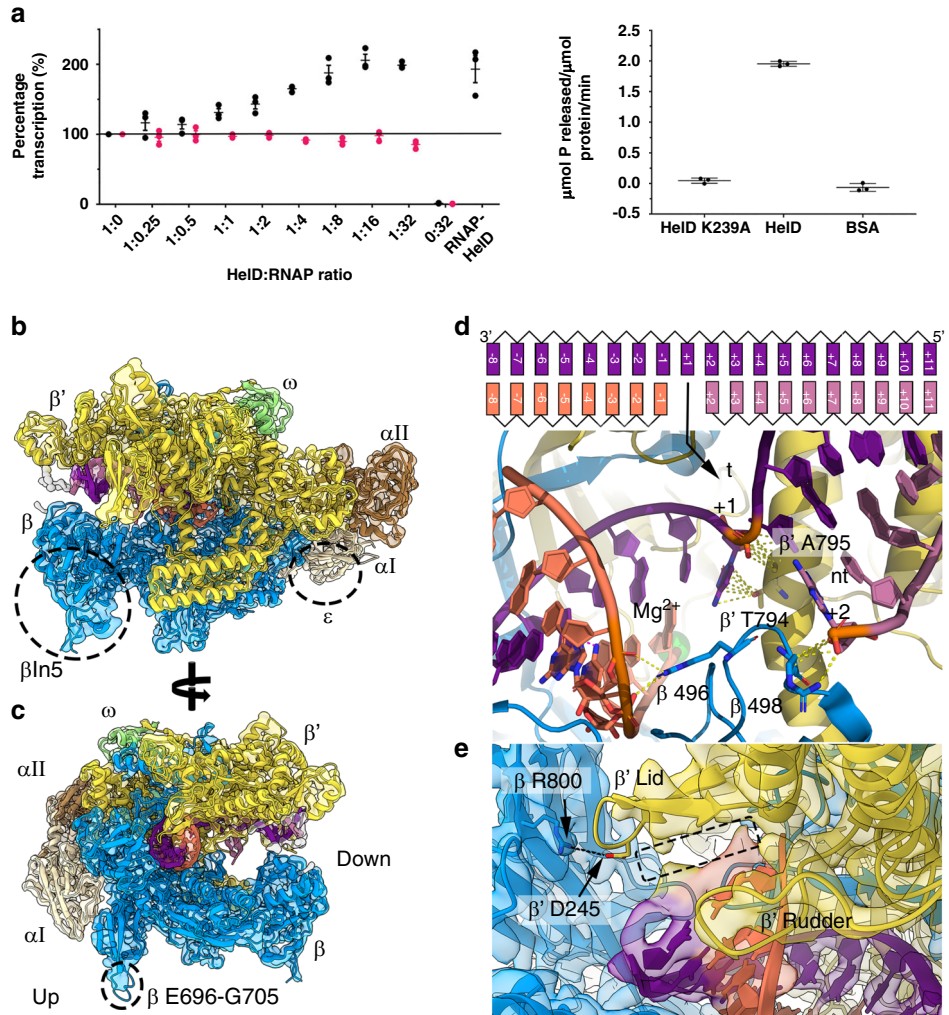

**Fig. 1 Structure of RNAP elongation complex. a** Left-hand graph shows multi-round transcription assays of purified RNAP core supplemented with increasing amounts of HelD (as a ratio with RNAP), HelD only (0:32, negative control) and the native RNAP-HelD complex purified from *B. subtilis* (RNAP-HelD). Black dots represent data obtained using wild-type HelD, magenta dots, data obtained using HelD K239A. The right-hand graph shows ATPase assays of wild-type HelD, HelD K239A, and BSA (negative control). Error bars, SD. Each experiment was performed three times in technical duplicate. Source data are provided in the Source Data files. **b**, **c** Cryo-EM reconstruction of the RNAP elongation complex (EC). The electron density map is semi-transparent in the same colours as the subunits in the cartoon structure: αI beige, αII brown, β azure, β' yellow, ω light green, template DNA purple, non-template DNA pink, and RNA orange. The dotted circles show the location of the βIn5 insertion, ε binding site (**b**) and β E696-G705 insertion (**c**), respectively. The upstream and downstream sides of RNAP are indicated in **c**. **d** Top shows a schematic of the nucleic acids coloured as in panels **b** and **c**. +1 represents the template DNA nucleotide positioned within the active site. + integers represent nucleotides base paired as DNA on the downstream side of RNAP, and −ve integers represent the DNA-RNA hybrid on the upstream side. The arrow indicates the position of the unpaired +1 nucleotide in the active site of RNAP in the lower part of the panel. Intermolecular bonds (polar and non-polar) are indicated by the dashed lines. Relevant template (t) and non-template (nt) nucleotides are numbered appropriately and amino acids shown with the RNAP subunit as a prefix. The catalytic $Mg^{2+}$ ion is shown as a green sphere to indicate the active site. **e** An enlarged view of the upstream side of the EC close to the entry of the RNA exit channel. The salt bridge formed between β R800 and β' D245 is shown as black dashed line. The space below the β' lid that could accommodate a $9^{th}$ DNA-RNA base pair is shown as a dotted rectangle, with the β' rudder helping guide RNA towards the β' lid and the RNA exit channel.

the *T. thermophilus* EC structure shows steric clashes between ε and the insertions (Supplementary Fig. 4c) raising the possibility that they serve similar functions. The location of ε also corresponds to that of a domain of archaeal and eukaryotic Rpo3/RPB3 subunits associated with enzyme stability (boxed insert, Supplementary Fig. 4a) and it is interesting to note that both *B. subtilis* (able to grow up to ~52 °C) and the thermophile *T. thermophilus* (up to ~79 °C) have structural elements/subunits located in this area that links the $α_2$, β, and β' subunits whereas the mesophilic *E. coli* and *M. smegmatis* do not.

The ω subunit in *B. subtilis* is 67 amino acids *vs* the 80 amino acid length of *E. coli* ω. The main structural difference appears to

be in the lack of a C-terminal α-helix which is prominent in *E. coli* RNAP, but lacking in *B. subtilis* and *Mycobacterial* structures. As with all other RNAP core and EC structures solved to date, the C-terminal domains of the α subunits were not visible due to the flexible linker connecting the N- and C-terminal domains.

Detailed examination of the elongation complex also revealed important features associated with mechanistic aspects of RNA synthesis. The density for fork-loop 2 (FL2) is well defined, consistent with its role in DNA strand separation on the downstream edge of the transcription bubble. The EC active site is similar in structure to that reported previously for the *T. thermophilus* and *E. coli* ECs[22,24] (2O5I, and 6ALF, respectively)

**Table 1 Cryo-EM data collection, refinement and validation statistics.**

| | RNAP elongation complex (EMD-21920, PDB 6WVJ) | RNAP-HelD complex (EMD-21921, PDB 6WVK) |
|---|---|---|
| **Data collection and processing** | | |
| Molecular mass (kDa) | 344.200 | 442.060 |
| Magnification | 59,524 | 59,524 |
| Voltage (kV) | 300 | 300 |
| Electron exposure (e−Å−2) | 52.2 | 63.6 |
| Defocus range (μm) | 0.6–2.8 | 0.6–2.8 |
| Pixel size (Å) | 0.84 | 0.84 |
| Symmetry imposed | C1 | C1 |
| Initial particle images (no.) | 1069336 | 580,468 |
| Final particle images (no.) | 58,854 | 65,356 |
| Relative abundance (%) | 5.5% | 11.2% |
| Map resolution (Å) | 3.36 | 3.36 |
| FSC threshold | 0.143 | 0.143 |
| Dimensions (Length × width × hight in Å) | 150 × 112 × 123 | 156 × 154 × 139 |
| **Refinement** | | |
| Initial model used | 6WVK | 4NJC (ε) |
| Model resolution (Å) | 3.38 | 3.27 |
| FSC threshold | 0.5 | 0.5 |
| **Model composition** | | |
| Non-hydrogen atoms | 22367 | 28225 |
| Protein residues | 2802 | 3631 |
| Nucleic acid residues | 37 | — |
| Ligands | ZN: 2, MG: 1 | ZN: 2, MG: 1 |
| **B Factors (Å2)** | | |
| Protein | 63.03 | 46.39 |
| Nucleic | 111.00 | — |
| Ligand | 86.05 | 68.02 |
| **r.m.s deviations** | | |
| Bond lengths (Å) | 0.005 | 0.005 |
| Bond angles (°) | 0.692 | 0.774 |
| **Validation** | | |
| MolProbity score | 2.61 | 2.69 |
| Clashscore | 12.44 | 10.95 |
| Poor rotamers (%) | 5.49 | 6.70 |
| **Ramachandran plot** | | |
| Favoured (%) | 93.14 | 91.42 |
| Allowed (%) | 6.75 | 8.39 |
| Disallowed (%) | 0.11 | 0.19 |

and is in a post-translocation conformation with the 3′ end of the RNA transcript adjacent to the +1 site, with an unbent bridge-helix (BH) and the trigger-loop (TL) in the open conformation (Fig. 1d). This conformation is consistent with an elongation complex primed to receive an incoming NTP via the secondary channel.

FL2 residue β R498 interacts with the ribose and phosphate moieties of the final base in the non-template DNA strand prior to strand separation and formation of the transcription bubble and likely acts to facilitate formation of the downstream edge of the transcription bubble (Fig. 1d). The template base in the +1 site is held in position for base-pairing with the incoming substrate NTP through interaction with the highly conserved T794 and A795 of the BH, and may also be stabilised through stacking with the base in the −1 position (Fig. 1d). β R496 of FL2 interacts with the phosphodiester backbone of RNA bases 4 and 5 of the new transcript (Fig. 1d). In addition, residues Q469, P520, E521, N524, I528, K924 and K932 of the rifampicin binding

pocket of the β subunit form numerous interactions with the newly formed transcript (RNA residues 1–5) as has been previously reported[32,33]. The salt bridge between β R800 and β′ D245 that closes the primary channel off from the RNA exit channel[34,35] is clearly visible confirming that the elements on the upstream side of the transcription bubble, the rudder and lid, that are responsible for facilitating reannealing of the template and non-template strands and guiding RNA into the exit channel are in positions consistent with these assigned roles (Fig. 1e).

Electron density for RNA beyond the 8th nucleotide is poor, preventing further mapping of the transcript up to and through the exit channel. Likewise, density for DNA on the upstream side is poorly defined consistent with conformational flexibility in this region of RNAP[26]. Structural modelling, and comparison with ECs from other organisms[22,24], is consistent with there being sufficient space for a transcription bubble comprising a 9 bp template DNA-RNA hybrid prior to upstream DNA strand re-annealment and entry of the transcript into the exit channel guided by hydrophobic interaction with conserved β′ lid residues V242 and L244 (dotted box, Fig. 1e). The 9th RNA-DNA base pair has likely been degraded by nuclease activity during preparation of the complex. Overall, this structure serves as a valuable resource for structure-function studies with RNAP from the *Firmicutes* as well as being a reference structure to enable full understanding of the conformational changes involved in transcription complex recycling induced upon binding to HelD (below).

**The structure of an RNAP-HelD transcription recycling complex.** RNAP-HelD complexes were isolated from a culture of *B. subtilis* carrying a deletion of the *rpoE* gene that encodes the δ subunit, shown previously to act synergistically with HelD[12] (Supplementary Fig. 1). HelD itself is required for transcription complex recycling, and can perform this function independently of δ[12] which is absent in many organisms that contain genes encoding HelD proteins (*e.g. Clostridia*). The purified complex stimulated transcription ~2-fold, similar to that observed with in vitro assembled complexes[12], establishing its biological activity (Fig. 1a).

We determined the structure of the RNAP–HelD complex using single particle cryo-electron microscopy (cryo-EM) to 3.36 Å resolution (Supplementary Fig. 3), followed by atomic modelling (Fig. 2a, Table 1; Supplementary Movie 2). The resulting structure revealed that HelD, which is located on the downstream side of RNAP, has two arm domains that penetrate deep into the primary and secondary channels of RNAP (clamp arm; CA, and secondary channel arm; SCA, respectively, Fig. 2a–c), which account for the strong HelD-RNAP interaction[12,13,36]. The native RNAP-HelD preparation also contained the RNAP ε subunit[19] and showed it bound on the downstream side of RNAP in a concave space between the two α, β, and β′ subunits (Fig. 2a; see Supplementary Fig. 4).

HelD itself has an unusual 4-domain structure (Fig. 3a, b). The first 203 amino acids (aa) form the secondary channel arm (SCA), which is joined to a super-family 1 (SF1) 1 A domain (aa 204–291 and 539–610). In SF1 helicases, domain 1 A is split by the insertion of a 1B domain associated with helicase function[37], but in HelD it is split by the clamp arm (CA; aa 292–538). Residues 610–774 form a continuous SF1 2 A domain, which is usually split by a 2B insertion in SF1 helicases, that represents the 'head' of HelD. The overall appearance of the protein is that of a torso and head (domains 1 A and 2 A, respectively) flanked by a pair of muscular arms (SCA and CA), giving it a rather thuggish appearance (Fig. 3b, c).

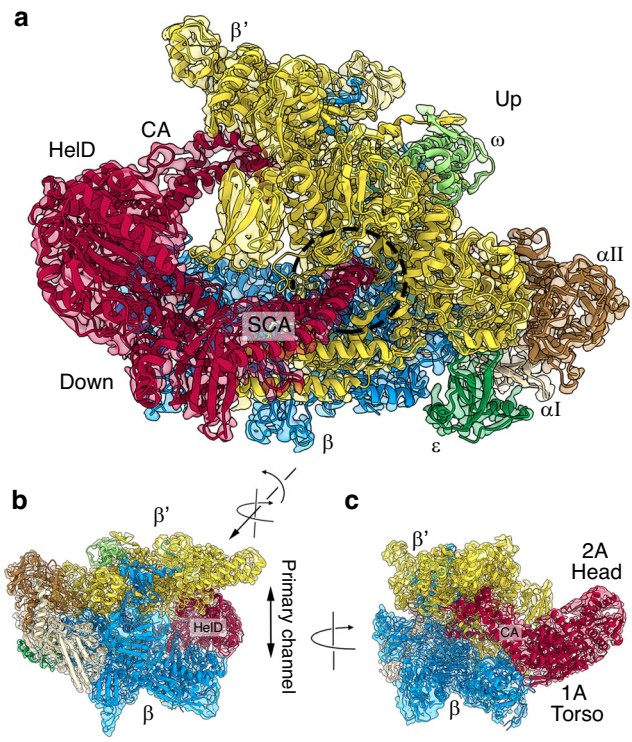

**Fig. 2 Structure of RNAP in complex with HelD. a–c** Different views of the cryo-EM reconstruction of the RNAP-HelD complex. The electron density map is semi-transparent in the same colours as in Fig. 1 with the addition of ε green, and HelD red. The HelD clamp arm (CA), secondary channel arm (SCA), 1 A Torso, and 2 A Head domains are labelled, as are the up- and downstream sides of RNAP (**a**). The primary channel that is formed between the β and β' subunits is shown in **b**, and the secondary channel indicated by the dotted circle in **a**. The change in orientation between the views in panels **a** and **b** is indicated by the arrows, with the right side arrow indicating a rotation of the view in **a** upwards from the α₂/ε end that would move the β' end down. The left arrow indicates the subsequent 165° clockwise rotation of the complex to give the view in **b**. The view in **c** is a simple 90° rotation of the orientation in **b**.

Although HelD is widely distributed amongst Gram-positive bacteria, the distinctive arm domains represent the regions of lowest sequence conservation despite being responsible for the majority of interactions with RNAP as well as for its transcription recycling activity[13] (Fig. 3a, Supplementary Figs. 5–7). It is also clear that there are at least two distinct classes of HelD (Classes I and II, Supplementary Figs. 5, 6, Supplementary Table 2); Class I is represented by the *B. subtilis* protein and is present in the low G + C Gram positives, whereas Class II is represented by the *M. smegmatis* protein (see accompanying paper by Kouba et al.[38]), and is present in the high G + C Gram-positives. Some organisms contain multiple copies of HelD (*e.g. Lactobacillus plantarum*, Class I; *Nonomuraea wenchangensis*, Class II; Supplementary Fig. 5), and even within the same organism, sequence conservation between the copies is relatively low in the SCA and CA domains (Supplementary Fig. 8). Previous studies showed that HelD in which the SCA (aa 1–203) had been deleted was still capable of binding RNAP, hydrolysing ATP, and binding DNA, but not transcription recycling[13]. These observations suggest that the function of the arm domains is centred around mechanical work rather than the formation of highly-conserved functionally-significant interprotein interactions.

Despite the clear separation into two classes, sequence alignment allowed the identification of conserved motifs common to all HelD proteins (Fig. 3a; Supplementary Table 2). The transcription recycling function of HelD is dependent on ATP hydrolysis[12], with ATP-binding motifs located in the 1 A (torso) domain (cyan residues, Fig. 3a, b). Alteration of the absolutely conserved K239 to A in the Walker A motif resulted in the complete loss of transcription recycling and ATPase activity (Fig. 1a). The remaining conserved motifs form a network of interactions that are mainly centred in the region between the SCA and 1 A domains, with the absolutely conserved residue W137 in a hydrophobic pocket between them (purple residues, Fig. 3a, c). These extensive interactions anchor the SCA to the 1 A domain, helping to couple ATP hydrolysis to mechanical movement of the CA (see below).

**HelD causes major conformational changes in RNAP.** Comparison of the core elements of the EC and RNAP-HelD structures (α₂ββ'ω subunits) shows HelD causes a major conformational change mainly due to the opening of the β' clamp by the CA, with very little change elsewhere (Fig. 4a, b; Supplementary Movie 3, and see below). PISA[39] was used to analyse protein-protein contacts in the RNAP-HelD and elongation complexes (Supplementary Table 3). Complexation with HelD reduces the contact area between RNAP subunits β and β' by over 6% while other contact areas remain similar, consistent with the extensive conformational change caused to the EC upon binding of HelD.

As part of transcription complex recycling, the elongating RNA as well as the DNA template needs to dissociate from RNAP. RNA passes through the exit channel on the upstream side of RNAP. There was no major conformational change to elements at the entry of the exit channel other than those that are translocated as part of the opening of the β' clamp (Fig. 4). The translocation of the β' clamp results in breaking of the conserved salt bridge between β R800 and β' D245 that is important in guiding RNA into the exit channel[34,35], increasing the width of the aperture from 11 to 20 Å (aₑ–aₑ; Fig. 4c, Supplementary Movie 3). This separation, along with widening of the primary channel, facilitates RNA exit from the complex.

The most dramatic effect of HelD on RNAP is the widening of the primary channel from 21 to 47 Å between β2 lobe P242 and β' clamp helix N283, that would cause a loss of contact with DNA in the primary channel, enabling recycling of RNAP (Fig. 4b, c). This is facilitated by the proximity of the CA to the SW5 region of the β' clamp, that acts as a hinge during clamp movement[18,40] (Supplementary Fig. 9, Movie 3).

Detailed examination of the SCA and CA interactions with RNAP enabled us to define the molecular events that occur during transcription complex recycling. Images of the active site region in the EC (Fig. 5a), RNAP-HelD complex (Fig. 5b) and an overlay of the two views (Fig. 5c) shows how HelD SCA insertion via the secondary channel causes distortion of the bridge-helix and trigger-loop as well as steric clashes with nucleic acids. The prokaryotic Gre factors, DksA, and eukaryotic TFIIS are known to bind in the secondary channel of RNAP via a pair of anti-parallel α helices/hairpin loop[41–43]. The acidic tips of these proteins reside close to, but on the downstream side of the catalytic Mg²⁺. The SCA of HelD bears superficial similarity to GreB/DskA, but is longer and the tip extends past the catalytic Mg²⁺ (Supplementary Fig. 10). The acidic tip (D56 and D57) will electrostatically repel the transcript upon penetration of the SCA into the active site, with the SCA causing significant steric clashes with the transcript and template DNA strand when fully inserted (Fig. 5c). The bridge-helix and trigger-loop, are dynamic structures that play a key role in the transcription cycle[44]; the entry of the SCA into the secondary channel causes partial folding of the open trigger-loop conformation observed in the EC

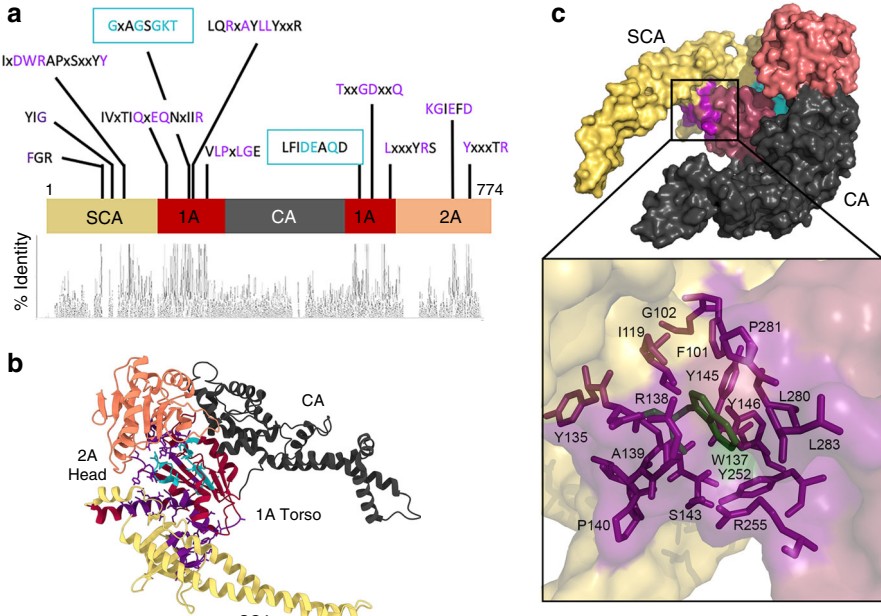

**Fig. 3 Structure and sequence conservation of HelD. a** Linear representation of the structure of HelD with a map of coloured domains and conserved sequence motifs (top), and a histogram of sequence identity (bottom). **b** The structure of HelD (bottom) coloured according to the schematic in panel **a**. SCA is shown in yellow, SF1 helicase-like domain 1 A in red, CA in dark grey, and SF1 helicase-like domain 2 A in orange. Conserved sequence motifs are shown in purple, with the Walker A/B ATP-binding site in cyan. **c**, shows a surface-rendered impression of HelD with the same colouring as in **b**. The bottom panel shows an expanded view of the boxed region in the top panel with a semi-transparent surface and conserved amino acids that form the 'Trp cage' shown as purple sticks, and the conserved Trp in dark green.

structure and a major distortion of the bridge-helix that would sterically clash with the template DNA in the active site (Fig. 5a–c; Supplementary Movie 3). Thus, the SCA tip itself, in combination with the distortion its insertion causes in the bridge-helix, will result in physical displacement of template DNA and RNA from the active site of RNAP, facilitated by electrostatic repulsion between the acidic SCA tip residues and the transcript.

The fully inserted tip of the SCA is in close proximity to the absolutely conserved active-site $\beta'_{447}$NADFDGD$_{453}$, forming a network of interactions around this motif, but does not directly interact with either the catalytic $Mg^{2+}$ or the Asp residues that coordinate it (Fig. 5d, Supplementary Table 4). Thus, upon dissociation of HelD, the core RNAP would be competent for re-use in transcription, as seen in the transcription recycling assays in Fig. 1a. Finally, insertion of the SCA into the secondary channel would block NTP entry into the active site.

The salt-bridge and H-bond contacts the CA makes with the $\beta'$ clamp are listed in Supplementary Table 4, but the bulk of interactions are made by hydrophobic residues with little sequence conservation between even closely-related genera (Supplementary Table 5, Supplementary Fig. 6a). This region is the location of an insertion that spans across the primary channel towards the active site in Class II HelD proteins (see accompanying paper by Kouba et al.[38]; Supplementary Fig. 6b). The tip of this insertion has a similar location to the tip of the SCA of *B. subtilis* Class I HelD and is also acidic, suggesting electrostatic repulsion of nucleic acids is also important in the activity of Class II HelDs. In our Class I HelD structure, the site of this insertion is close to an area of density in the cryo-EM reconstruction that at low threshold values could be consistent with the presence of nucleic acid (Supplementary Fig. 11). Examination of the surface charge of HelD revealed a region of high overall positive charge on the inside of the CA. Superposition with the nucleic acids from the EC show that this positively-charged patch is in a position where it

could interact with the downstream dsDNA (Supplementary Fig. 11a), consistent with nucleic acid-binding data[13]. It is also possible that this patch may be important for interaction with the unstructured negatively-charged C-terminal domain of δ which acts synergistically with HelD during transcription complex recycling[12] (see accompanying paper by Pei et al.[31]). The end of the CA forms a relatively flat ~320 Å$^2$ surface that acts as a platform to push up against the $\beta'$ clamp, resulting in loss of contact with the DNA bound in the EC (Figs. 2a–c, 4, Supplementary Movie 3). Therefore, the purpose of the CA appears to involve the opening of the $\beta'$ clamp through brute force rather than by the formation of a specific network of conserved interactions.

**Movement of the clamp arm of HelD drives conformational change in RNAP.** Closer examination of the RNAP–HelD complex using 3D variability analysis (3DVA)[45] allowed identification of regions of conformational flexibility that underpin the dynamic processes of HelD activity in transcription recycling. Overall, the region behind SW5 towards the α dimer, including the secondary channel and SCA of HelD, showed little or no conformational variability, but the primary channel encompassing elements of the β1 and 2 lobes and the $\beta'$ clamp did (Fig. 6a, Supplementary Movie 4).

The 3D variability analysis indicates that HelD causes the $\beta'$ clamp to open and twist so that the downstream side of RNAP opens slightly (curved cyan arrow, Fig. 6a). At the same time, the β2 lobe moves up (straight cyan arrow, Fig. 6a) along with a slight twisting of the β1 lobe and β flap (Fig. 6a, Supplementary Movie 4). With respect to HelD, there was no change in the SCA tip adjacent to the RNAP active site, but there was lateral movement of the portion located outside the secondary channel towards the $\beta'$ jaw (Fig. 6b). This resulted in little, if any, conformational change in the hydrophobic 'cage' surrounding the conserved W137 residue. Accordingly, there was relatively little

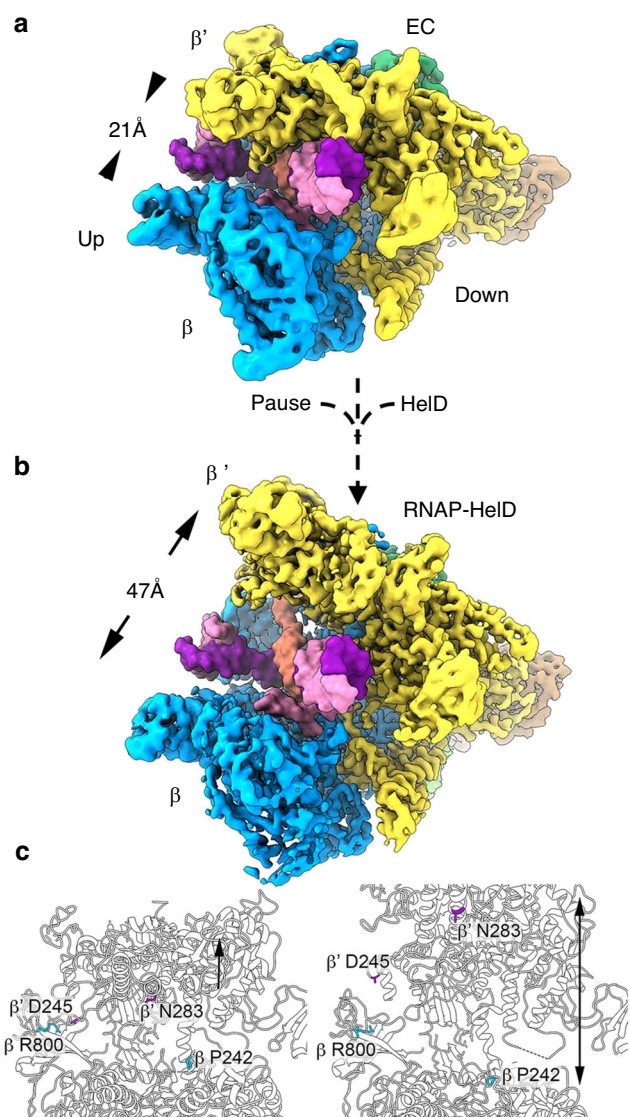

**Fig. 4 HelD-induced conformational change in RNAP. a**, **b** Cryo-EM electron density maps ($\alpha_2\beta\beta'\omega$ subunits only) of RNAP EC (**a**) and RNAP–HelD complex (**b**). HelD density has been removed for clarity, and nucleic acids from the *E. coli* EC (PDB ID 6ALF) superimposed over the *B. subtilis* EC nucleic acids as a visual aid for the scale of conformational changes that occur in the primary DNA-binding and RNA exit channels upon binding HelD. Nucleic acids are coloured the same as in Fig. 1. The longer RNA from the *E. coli* EC helps illustrate the increase in aperture of the RNA exit channel. **c** left side EC, right side RNAP-HelD complex with HelD removed for clarity, with β (cyan) and β′ (purple) residues shown to illustrate the change in aperture of the RNA exit channel β R800 and β′D245 and DNA binding channel β P242 and β′N283. Black arrow in left panel indicates the region of the β′ clamp that is contacted by the HelD clamp arm and the double-ended arrow in the right panel indicates the movement of subunits away from each other on HelD binding.

change in the torso (1 A) domain and ATP-binding site, but the head (2 A) domain moved away from the downstream side of RNAP (curved and straight orange arrows Fig. 6a, b, respectively). The CA of HelD rises up and out slightly, causing the upward twist on the downstream side of the β′ clamp (orange arrow, Fig. 6a; Supplementary Movie 4). Therefore, the results of the 3D variability analysis are consistent with the SCA acting as a wedge that permits conformational change through movement of the CA. The CA is located in a position equivalent to an SF1 helicase

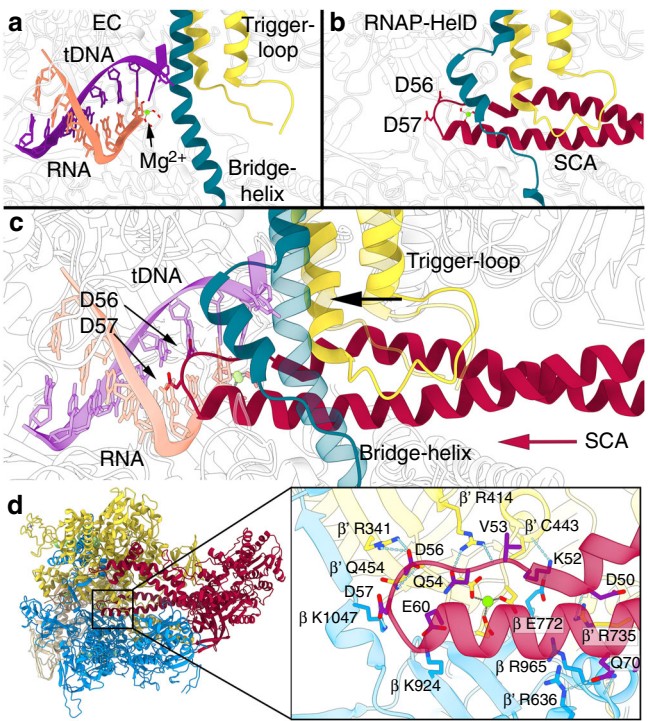

**Fig. 5 HelD interactions with RNAP. a** View of the active site region of the EC with the bridge-helix in teal, trigger-loop in yellow, template DNA strand in purple and RNA in orange. **b** The same view of the active site region of the RNAP-HelD complex with the SCA of HelD shown in red with the acidic D56 and D57 residues shown as sticks. **c** An overlay of the regions shown in **a** and **b** with the insertion of the SCA into the secondary channel indicated by the red arrow. EC elements are shown semi-transparent with nucleic acids coloured according to the scheme in Fig. 1. The bridge-helix and trigger-loop elements that are distorted by the SCA are shown, with the direction of movement from EC to HelD complex indicated by the black arrow. The catalytic $Mg^{2+}$ ion is shown as a green sphere. **d** Hydrogen-bond and salt-bridge interactions (dashed blue lines) between the tip of the SCA (red semi-transparent cartoon with purple sticks) and RNAP. Subunit colouring is the same as in Fig. 1. The conserved active site Asp residues that chelate the catalytic $Mg^{2+}$ (green sphere, grey dashed lines) are also shown to illustrate how the tip of the SCA cages but does not interact directly with residues involved in RNAP catalysis. The enlarged box on the right corresponds to the boxed region shown for the whole complex on the left.

1B domain that utilises ATP hydrolysis to undergo conformational changes required for helicase/translocase activity[46,47] and ATP binding/hydrolysis is required for release of HelD (see accompanying papers by Kouba et al. and Pei et al.[31,38]), most likely due to movement of the CA arm, consistent with the observed conformational flexibility in this region.

In our structure, and those of the accompanying papers by Pei et al. and Kouba et al., no density for any NTP could be detected in the ATP binding site, even on addition of ATP or non-hydrolysable analogues. In order to bind ATP, domains 1 A (torso) and 2 A (head) need to rotationally open as observed for SF1 helicases[48]. Our 3DVA suggests this is most likely via movement of the 2 A (head) domain (Fig. 6a, b). However, the sequence from F183-G190 linking the SCA to the 1 A (torso) domain sterically blocks access to the ATP binding site and this 'gate' region will also need to open to allow ATP binding and subsequent ADP release (Supplementary Fig. 12). There is no intramolecular bonding between residues T185-I189 and either the SCA or IA (torso) domain, and this may provide the

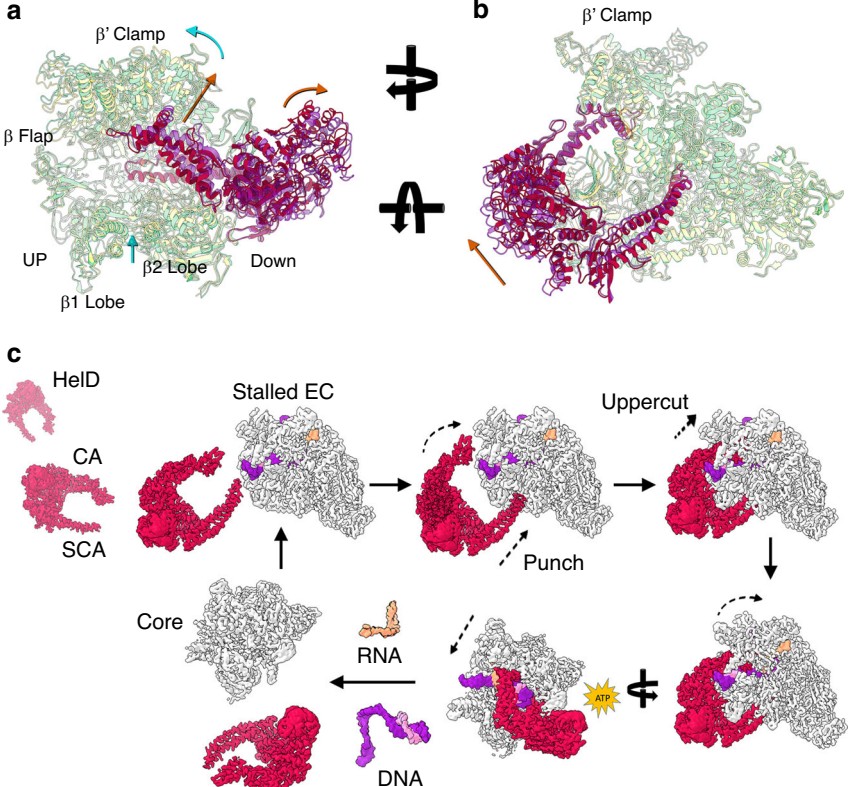

**Fig. 6 3D variability analysis of the RNAP–HelD complex and a model for HelD-catalysed RNAP recycling. a**, **b** Different orientations of the most distinct conformations determined by 3DVA. HelD is shown in solid colours with RNAP semi-transparent. The red conformation of HelD is matched to the pale green conformation of RNAP, and the purple HelD with the yellow RNAP. Orange arrows indicate the movement of the juxtaposed region of HelD, and cyan arrows regions of RNAP. Structural elements of RNAP referred to in the text are labelled as well as the up- (UP) and downstream (DOWN) sides of RNAP. **c** Model for HelD-catalysed recycling. HelD is shown in red, RNAP in white, DNA in purple (template) and pink (non-template), and RNA in orange. Clockwise from top left; HelD locates a stalled EC and binds with the SCA penetrating the secondary channel and the CA moving into position on the β′ clamp. The SCA is wedged deep within RNAP, and through conserved interactions with the 1 A torso domain, locks HelD in position. The contacts the CA makes with the β′ clamp, open the DNA-binding and RNA-exit channels to enable dissociation of nucleic acids from the EC, possibly assisted by interaction of the DNA with the positively-charged patch on the CA of HelD. HelD (and nucleic acid) dissociation is facilitated by the conformational changes facilitated by ATP binding/hydrolysis (ATP flash). Finally, core RNAP that has been released from the complex is free to enter a fresh round of transcription.

necessary flexibility for gate opening and closing. Given that the ATP binding site was not accessible in all of the structures that are forcing open the DNA binding clamp of RNAP, gate opening may be an event that occurs on conformational change of the CA during nucleic acid release and RNAP recycling.

## Discussion

These results provide the foundations for a model of HelD catalysed transcription recycling (Fig. 6c). HelD is present at low intracellular levels compared to RNAP[12,49] and this may help restrict it to targeting ECs that have entered a long-term pause[11]. The SCA penetrates the secondary channel (Punch), displacing the transcript and template DNA, blocking the catalytic $Mg^{2+}$ and NTP entry. Through conserved inter-domain interactions with the torso, the SCA anchors HelD on RNAP. HelD then forces the primary channel open through interaction of the CA with the β′ clamp (Uppercut). The action of both arms serves to displace nucleic acids from the active site, through separation of the β-flap and β′-clamp by ~10 Å, and opening the primary channel by ~36 Å. Dissociation of HelD follows conformational change and ATP binding/hydrolysis (see accompanying papers by Kouba et al. and Pei et al.[31,38]) consistent with SF1 helicase dynamics that are transmitted to the CA[46,47], closing of the

primary channel, and recycling of RNAP. The conservation of HelD across the Gram positive bacteria indicates this previously unknown mechanism for transcription complex recycling is of considerable importance. Determination of the precise molecular details by which highly diverged structures perform this role represents an exciting new avenue of research.

## Methods

**Strains**. *E. coli* BL21 (DE3) was used for overproduction of core *B. subtilis* RNAP, HelD and σ[A]. RNAP holoenzyme (HE) and HelD complexes were purified from *B. subtilis* LK637[50] carrying a deletion to the *rpoE* gene encoding δ, and a 3′ *his* tag on the *rpoC* gene to facilitate purification.

**Plasmids**. Recombinant *B. subtilis* RNAP core (α₂ββ′ω) has previously been overproduced using a two plasmid system[20] although the use of two different plasmids could result in poor yields if one plasmid was present at lower levels than the other during overproduction. To make the process more efficient a single plasmid system was constructed. pNG219[20] containing *rpoA*, *rpoB*, and *rpoC* was linearised with NotI, purified and used in a Gibson assembly reaction with a 356 bp gBlock® (IDT, Singapore) construct comprising 40 bp 5′ and 3′ homology to the linearised pNG219 DNA flanking an additional phage T7 promoter, NcoI and XbaI restriction sites, a ribosome binding site and the *rpoZ* gene. The resulting plasmid was named pNG1256 (sequence and plasmid DNA available through Addgene, ID 149710).

The HelD K239A mutant was constructed by PCR mutagenesis[51] of the pHelD-His6 plasmid[12]. The PCR contained 1X NEB Q5 reaction buffer (B9027), 200 μM dNTPs, 0.5 μM forward primer 5′ GCGGGGGCAACATCGGCCGCGCTTCAG 3′,

0.5 μM reverse primer 5′ ATGTTGCCCCGCTGCCAGCCGCTCCC 3′, 10 ng pHelD-His, 0.25 μl Q5 DNA polymerase (NEB) and sterile H$_2$O up to 25 μl total reaction volume. Thermocycling conditions were initiated at 98 °C for 3 min, followed by denaturation at 98 °C for 10 s, annealing at 69 °C for 30 s, extension at 72 °C for 4.5 min for 12 cycles, then a single cycle of annealing at 59 °C for 30 s and a final extension at 72 °C for 30 min. Plasmid sequence (pNG1304) was confirmed by Sanger sequencing (AGRF). Overproduction and purification of the protein was the same as for the native HelD (below).

Plasmid template for the MGA transcription assay consists of a construct containing three strong consensus promoters, *Thermus* VV1-2/D2[52], pGP31 from *Bacillus* phage SPO1 and LacUV5 that directed transcription of an array containing 12 direct repeats of the MGA sequence (5′-GGATCCCGACTGGCGAGAGCCAGG TAACGAATGGATCCTAAAAAC-3′) followed by an *E. coli* tRNA-trp terminator. This construct was synthesised by GenScript and cloned into the EcoRV site of pUC57-Simple to give pNG1299, (sequence and plasmid available through Addgene, ID 149709). pNG1299 was propagated in *E. coli* NEB® Stable (C3040H, New England Biolabs Inc) to avoid recombination of the MGA array. Supercoiled plasmid template was prepared as described by[53] using the reagents from an ISOLATE II Plasmid Mini Kit (BIO-52057, Bioline).

**B. subtilis EC RNAP purification**. A seed culture (40 ml) of *E. coli* BL21(DE3) transformed with pNG1256 was grown in LB supplemented with 100 μg/ml ampicillin at 37 °C to an A$_{600}$ of 0.5 and was used to inoculate 4 L of auto-induction medium[54] supplemented with 100 μg/ml ampicillin. The culture was grown at 30 °C for 30 h, cells harvested by centrifugation at 4,000 × g, 4 °C, 20 min, and washed pellets stored at −80 °C. The frozen cell pellet was resuspended in 100 ml HisA buffer (20 mM KH$_2$PO$_4$ pH7.8, 500 mM NaCl, 20 mM imidazole) supplemented with EDTA-free protease inhibitor cocktail (1 × concentration S8830, Sigma-Aldrich) and 100 μl of 4 mg/ml DNaseI (DN25, Sigma-Aldrich) at 4 °C. Cells were lysed by repeated passage through an Avestin C5 homogeniser at ~20 kPa, and the lysate clarified by centrifugation at 16,000 × g, 4 °C, 20 min.

The resulting supernatant was loaded onto a 5 ml HisTrap FF column pre-equilibrated with HisA buffer. The column was washed with 4% HisB (HisA + 500 mM imidazole), and RNAP eluted with 50% HisB. Following dialysis in QA buffer (20 mM Tris-HCl pH 7.8, 150 mM NaCl, 10 mM MgCl$_2$, 1 mM DTT) the sample was loaded onto a 1 ml MonoQ column pre-equilibrated in TrisA. A gradient of 0–50% TrisA supplemented with 1 M NaCl over 10 ml was used to elute proteins with RNAP core eluting as a peak at ~0.35 M NaCl.

Purified RNAP was dialysed into QA buffer (20 mM Tris-HCl pH 7.8, 150 mM NaCl, 10 mM MgCl$_2$, 1 mM DTT), concentrated with an Amicon® Ultra-15 Centrifugal Filter with a 3 KDa NMWCO (UFC900324, Merck Millipore) and small aliquots snap-frozen in N$_2$(l) and stored at −80 °C. Two different concentrations of purified RNAP were prepared at 23.3 mg/ml and 8.2 mg/ml for storage. Protein and nucleic acid content was determined with a Qubit fluorometer using the protein and HS DNA assays (Supplementary Table 1).

**Bacillus subtilis σ$^A$ purification**. σ$^A$ was overproduced and purified[25] with the following modifications; after HisTrap FF column purification, fractions containing σ$^A$ were dialysed overnight into 50 mM NaH$_2$PO$_4$ pH 8.0, 150 mM NaCl. Dialysate was loaded onto a Mono Q 5/50 GL column (17516601, GE Life Sciences) at a rate 0.5 ml/min, and washed with 50 mM NaH$_2$PO$_4$ pH 8.0, 150 mM NaCl for 10 ml at a flow rate of 0.5 ml/min. σ$^A$ was eluted with a gradient of 150–500 mM NaCl over 20 ml followed by a step to 1 M NaCl for 3 ml. Fractions containing σ$^A$ were dialysed into 50 mM NaH$_2$PO$_4$ pH 8.0, 150 mM NaCl and concentrated with an Amicon® Ultra-15 Centrifugal Filter with a 3 kDa NMWCO. Concentrated samples were snap-frozen in N$_2$ (l) and stored at −80 °C before use.

**B. subtilis HelD purification**. HelD and HelD K239A were overproduced and purified[12] with the following modifications; after HisTrap FF column purification fractions containing HelD were dialysed at 4 °C into (20 mM Tris-HCl pH7.8, 10 mM MgCl$_2$, 150 mM NaCl, 5 mM DTT) and applied to a 1 ml HiTrap Heparin HP column (17040601, GE Lifesciences) at a flow rate of 1 ml/min. The column was washed with 5 ml 20 mM Tris-HCl pH7.8, 10 mM MgCl$_2$, 150 mM NaCl) at 1 ml/min and eluted with a gradient of 150 mM-1000 mM NaCl over 20 ml. Fractions containing HelD were dialysed overnight into QA buffer and concentrated using an Amicon® Ultra-15 Centrifugal Filter with a 3 kDa NMWCO (UFC900324, Merck Millipore). The concentrated sample was snap-frozen in N$_2$ (l).

**B. subtilis HelD-RNAP complex**. *B. subtilis* LK637[50] (Δδ) was grown at 45 °C in LB in baffled flasks to maximise aeration to late exponential phase, cells pelleted by centrifugation, and washed pellets stored at −80 °C. Frozen pellets from 8 L culture were resuspended in 50 ml HisA buffer (above) supplemented with EDTA-free protease inhibitor cocktail (1.8× concentration S8830, Sigma-Aldrich) at 4 °C, and lysed by multiple passage through an Avestin C5 homogeniser at ~25 kPa, and the lysate clarified by centrifugation at 16,000 × g, 4 °C, 20 min.

RNAP was purified using a 5 ml HisTrap FF as above and the eluted sample dialysed overnight against QA buffer (20 mM Tris-HCl pH 7.8, 150 mM, 10 mM MgCl$_2$, 1 mM EDTA, 5 mM DTT) at 4 °C. Following clarification by centrifugation,

the overnight dialysate was loaded onto a 1 ml MonoQ column pre-equilibrated in QA buffer. RNAP was eluted using a 13.5 ml 150–500 mM NaCl gradient in QA buffer. RNAP containing fractions were pooled and dialysed overnight at 4 °C against QA buffer prior to loading onto a 1 ml HiTrap Heparin HP column pre-equilibrated in QA buffer. RNAP was eluted as two peaks using a 150–1000 mM NaCl gradient over 15 ml. The first peak corresponded to Holoenzyme (~650 mM NaCl), and the second, minor peak, to the HelD complex (~950 mM NaCl).

Holoenzyme and HelD complex fractions were dialysed separately against buffer QA overnight at 4 °C, concentrated and small aliquots snap-frozen in N$_2$(l) and stored at −80 °C. Holoenzyme fractions were 7.46 mg/ml and HelD complex fractions 2.57 mg/ml.

**Transcription assays**. An in vitro transcription assay adapted from[55] was used to assay the activities of RNAP and RNAP-HelD complexes. Briefly, 20 μl transcription reactions containing 80 nM of core RNAP, 240 nM of σ$^A$ and 0–2560 nM of HelD or HelD K239A in transcription buffer (40 mM Tris-HCl pH 7.5, 50 mM KCl, 10 mM MgCl$_2$ 0.02% (v/v) Triton-X100, 8 mM DTT) were assembled in a well of a black, half area, flat-bottomed, non-binding 96 well microplate (CLS3686, Corning) and were incubated for 15 min at 37 °C with shaking. To initiate the transcription reaction 20 μl of start solution containing 1 mM rNTPs (N0466S, New England Biolabs Inc) and 10 nM pNG1299, in transcription buffer was added. The plate was sealed with a clear adhesive plate seal (WHA-7704-0001, Whatman) and incubated with shaking at 37 °C for 15 min. Reactions were stopped by the addition of 40 μl of ice-cold stop solution (10 mM Tris-HCl pH8.0, 1 mM EDTA, 144 μM malachite green oxalate salt M6880, Sigma-Aldrich) and developed on ice for 5 min. The plate was read using a Pherastar FS (BMG Labtech) using a 610 nm excitation, 675 nm emission optical module. Percentage transcription was calculated relative to a 1:0 RNAP:HelD reaction. Each experiment was performed three times in technical duplicate.

**ATPase assays**. ATPase activity was determined by malachite green assay[56]. Malachite green reagent was prepared immediately prior to use by mixing 0.045% (w/v) malachite green with 5% (w/v) ammonium molybdate in 4 M H$_2$SO$_4$ in a 3:1 (v/v) ratio, and passing through a 0.45 μm filter. Reactions were carried out based on methods described in work[13] with the following modifications. Reaction mixtures contained 100 pmol of protein, 10 mM ATP, 20 mM Tris-HCl pH 7.8, 10 mM MgCl$_2$, and 150 mM NaCl in a final volume of 100 μl. All reaction components, with the exception of ATP, were assembled and incubated at 25 °C for 5 min to equilibrate. Following incubation, ATP was added to each mixture and the reaction was allowed to proceed for 30 min at 25 °C. Upon completion, 800 μl of malachite green reagent was added to each reaction, incubated for 1 min, followed by addition of 100 μl of 34% (w/v) sodium citrate. The colorimetric change was allowed to develop for 20 min at 25 °C, following which 250 μl of each reaction was transferred to a microplate and absorbance read at 660 nm on a Pherastar FS (BMG Labtech) plate reader. Phosphate released was quantified by comparison to a standard curve constructed from KH$_2$PO$_4$. Reactions were performed in technical duplicates and results averaged across 3 independent assay replicates.

**Preparation of cryo-EM grids and cryo-electron microscopy**. Between 2 and 2.5 μl of 2.57 mg/ml RNAP-HelD complex or 3.0 mg/ml EC diluted in QA buffer were deposited onto glow-discharged UltrAuFoil 1.2/1.3 or Quantifoil 1.2/1.3 cryo-electron microscopy grids and blotted for 5 s before plunge freezing into liquid ethane using a Mark IV Vitrobot (FEI). Data were collected on a Titan Krios (Thermo Fisher) electron microscope operated at 300 kV and equipped with a Gatan BioQuantum LS 967 energy filter and Gatan K2 Summit detector, operated in unfiltered mode. Data were collected in electron counting mode at a pixel size of 0.84 Å/pixel and a calibrated sample-to-pixel magnification of 59524 x, (microscope user interface listed magnification, 165000 × EFTEM). For the HelD complex, movies were collected as a series of 60 frames with a total accumulated dose of 63.6 e$^-$/Å$^2$. For the EC, movies were collected as a series of 40 frames and a total accumulated dose of 52.2 e$^-$/Å$^2$. For both datasets, data were collected using automated data collection in EPU, with a defocus range of 0.6–2.8 μm. Approximately 60% of the data for both the RNAP HelD complex and the EC were collected at 20° stage tilt to compensate for the effects of preferred particle orientation.

**Image processing of the HelD complex**. A total of 4331 images (1859 at 0° tilt, 2472 at 20° tilt) were collected for the RNAP-HelD complex. All image processing was performed in RELION 3.1[57] unless otherwise indicated. Movies were aligned and dose-weighted using MotionCor2[58] before contrast-transfer function (CTF) estimation was performed on the motion-corrected images using GCTF[59]. Particle picking was performed in CrYOLO[60] using a pre-trained general model which had been refined using a subset of manually picked training data. Particle coordinates were contrast-inverted, normalised and extracted in RELION. Following particle picking, a total of 580,468 particles were extracted and subjected to several rounds of 2D classification for initial cleaning of the particle data, which resulted in a subset of 379,179 particles. Subsequent rounds of 3D classification isolated a smaller subset of 65,356 particles. Due to moderate preferred orientation of the specimen, a large portion of the particles were excluded to minimise resolution

anisotropy in the final reconstruction. These particles were then subjected to iterative Bayesian polishing and CTF refinement with higher-order aberration correction in RELION until further processing ceased to yield an increase in resolution. Following refinement of the per-particle motion and CTF parameters, post-processing of the final reconstruction yielded a resolution of 3.36 Å as determined by the Gold-standard Fourier-Shell Correlation (FSC = 0.143) criterion in RELION[61]. The final refined map was then subjected to density modification and automated model-based based sharpening in Phenix[62,63] (see Supplementary Fig. 3).

Following reconstruction in RELION, the final subset of 65,356 particles was subjected to 3D variability analysis in Cryosparc[45]. 3D variability analysis was performed solving for 3 conformational modes, and the results visualised in ChimeraX[64].

**Image processing of the elongation complex**. A total of 5185 images (2955 at 0° tilt, 2230 at 20° tilt) were collected for the EC. All processing was performed as described for the HelD complex above unless otherwise indicated. Following particle picking, a total of 1,069,336 particles were extracted in RELION and subjected to several rounds of 2D classification to obtain a smaller subset of 355,795 particles. This subset was subjected to several rounds of 3D classification to remove incomplete particles and over-represented orientations to isolate a final subset of 58,854 particles. These particles were then subjected to Bayesian polishing, CTF-refinement and postprocessing as for the HelD complex above. Following postprocessing, the final reconstruction yielded a resolution of 3.36 Å as determined by the GSFSC criterion in RELION. As for the RNAP-HelD complex, the final refined map was then subjected to density modification and automated model-based based sharpening in Phenix[62,63] (see Supplementary Fig. 2).

**Structure building and refinement**. Initial model building for the core RNAP subunits commenced using a homology model generated previously[19], which was fitted into the density of the HelD complex using rigid body fitting in CHIMERA followed by molecular-dynamics flexible fitting in NAMD[65]. The model was subject to cycles of manual model building in COOT[66] followed by refinement by phenix.real_space_refine. De novo atomic modelling for the HelD subunit was performed by initial modelling of HelD in Phenix using phenix.map_to_model[62], followed by model building in COOT[66].

The ε subunit was modelled based on homology with a known structure from *Geobacillus stearothermophilus* (PDB ID: 4NJC)[67]. Refinement of the elongation complex commenced using the RNAP-HelD model, which was placed into density using phenix.dock_in_map, followed by cycles of model building in COOT and refinement in phenix.real_space_refine. Density-based sequence was built for nucleic acids. All models were further refined in ISOLDE[68] and phenix.real_space_refine until deemed final.

**Sequence analysis**. *B. subtilis* HelD sequence (UniProtKB - O32215) was used to identify similar proteins using the NCBI CDART search programme[69]. Sequences from diverse organisms were selected from the 24426 hits defined as HelD-related helicases for alignment using ClustalX 2.1[70]. Sequences were also selected from the DUF4968 domain-containing protein (74 hits) and the multispecies: DUF4968 domain-containing protein (37 hits) categories for characterisation of their HelD-like sequences. Phylogenetic trees were constructed using NCBI COBALT[71], and sequence conservation mapped to structure using ConSurf[72].

**Reporting summary**. Further information on research design is available in the Nature Research Reporting Summary linked to this article.

## Data availability

CryoEM maps have been deposited in the Electron Microscopy Data Bank (https://www.ebi.ac.uk/pdbe/emdb/) under accession codes EMD-21921 (RNAP-HelD) and EMD-21920 (RNAP elongation complex). Structure coordinates have been deposited in the RCSB Protein Data Bank (https://www.rcsb.org/) with accession codes 6WVK (RNAP-HelD) and 6WVJ (RNAP elongation complex). Plasmids pNG1256, pNG1299 and pNG1304 are available from Addgene (https://www.addgene.org) under accession numbers 149710, 149709, and 162488, respectively. Other data supporting the findings of this study are available from the corresponding authors on request. Source data are provided with this paper.

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

## Acknowledgements
We would like to thank Profs Nick Dixon and Rick Lewis for helpful comments on the manuscript, and members of our respective laboratories and those of Dr Libor Krasny and Prof Markus Wahl for discussions. P.J.L. acknowledges the assistance of Sarah Pichereau during the purification of RNAP-HelD complexes. This work was supported by grants from the Priority Research Centre for Drug Discovery, University of Newcastle (P.J.L.), NUW Alliance (G1801287 to P.J.L., A.J.O and G.T.), and NHMRC (GNT1184012 to G.T.). T.N., M.M., and C.J.D. were funded through PhD scholarships from the Australian Government.

## Author contributions
P.J.L., M.M. and C.J.D. cloned genes, produced proteins/complexes and performed experiments. T.N. prepared and imaged cryo-EM samples with S.H.J.B. and J.B., T.N. processed cryo-EM data with S.H.J.B, P.J.L. and G.T., and built atomic models with A.J.O. and G.T. All authors contributed to the analysis of the data and the interpretation of the results. P.J.L. wrote the manuscript with contributions from the other authors. P.J.L. and G.T. supervised work in their respective groups. P.J.L. conceived and coordinated the project.

## Competing interests
The authors declare no competing interests.
