## [Peer Review File · Nature Communications]

REVIEWER COMMENTS

Reviewer #1 (Remarks to the Author):

Newing et al., describe the first cryo-EM structures of *B. subtilis* RNAP in both EC form and bound to the reactivation factor HelD. The structures reveal novel aspects of Firmicutes transcription regulation and provide a mechanism for DNA release by HelD. The study is of general interest to the transcription community, however, a number of biochemical experiments are missing and are needed to confirm the structural results. Overall, the figures and movies are well rendered. The abstract and discussion could be improved by closing with an overall summary statement, and some phrasing in the paper can be improved. The reviewer recommends the manuscript for publication in Nature Communications if the authors can address the specific comments below.

Specific comments:

-Figures 1 and 2: the authors prepared the RNAP•HelD complex from *B. subtilis* for the experiments in Figure 2. The EC complex in figure 1 was purified by overexpression in *E. coli*. Given that the observed stimulation is far less for the proteins expressed in *E. coli*, the authors should comment on reasons for this observation (perhaps the presence of the epsilon subunit?). Additionally, the chromatogram provided in Extended Data Figure 1 indicates that the RNAP purified from *B. subtilis* is not contaminated with nucleic acids, whereas the overexpressed version clearly has nucleic acid contamination. The authors should provide a chromatogram of the *E. coli* expressed protein in addition to that provided for the protein purified from *B. subtilis*.

-Figure 1a/2a Provide a control with only HelD and no RNAP. This is important because malachite green can be used to detect free Pi in solution at low pH. The reviewer is aware that described assays were performed at physiological pH, but it is still critical to show that the observed effects are not due to HelD ATP hydrolysis activity.

-Figure 1,2,4 : Label upstream and downstream on RNAP.

-Methods- Authors refer to "gold standard FSC". They should state that this has a value of 0.143 and cite Rosenthal and Henderson, JMB 2003.

-Lines 227-229 and 311-313: The authors state that the HelD arm interactions with RNAP are mostly based on mechanical force. Given the stability of the complex in high salt buffers, there must be some functional residues or patches. It would greatly help if the authors mutated some of the surfaces, particularly salt bridges or hydrophobic patches, to show which residues are important for the observed association. Proteins do not bind each other randomly. It would also help if specific, relevant amino acid contacts are included in the text. The resolution of the structure should be sufficient to describe these interactions, and the authors have a compiled table that already lists some residues that may be important for the observed interaction.

-Biochemical assays to show that HelD ATPase activity is necessary for HelD recycling activity. The authors should mutate the HelD ATPase and see if the recycling activity is reduced or association with RNAP is affected. It appears that this may have been done in the accompanying Pei and Kouba manuscripts, but it would help if this data was included in this manuscript.

-Figure 5a: It assumed that this figure is an overlay of the EC structure with the HelD structure. The authors should state this in the figure legend. Additionally, it would help to have a side by side comparison of a normal EC active site versus HelD bound. The distortion in the bridge-helix is quite profound.

-Figure 5a: It would help to have an Extended Data figure with a comparison of the HelD positioning relative to GreB/DskA and TFIIS. All use acidic residues to reactivate transcription.

Minor concerns:

-several times the phrase (e.g. lines 23 and 61) "cryo-electron microscopy and single-particle analysis" is used. Generally, "single particle cryo-electron microscopy" is used in the field.

-Descriptors like "remarkable" (line 59-60) are a bit over the top for the introduction. It would be nicer to have a description of the HelD structure rather than using a platitude to describe the

protein. It would additionally help in the introduction to already indicate that HeID sits on the downstream side of RNAP.

-lines 42-44 somewhat redundant with the use of recycling. The last phrase "indicating it...recycling RNAP" can be left out

-line 112- show a supplementary figure with overlays of *B. subtilis* RNAP, *E. coli* RNAP, and *M. smegmatis*, etc. ECs. This would make the authors comparison more clear to the reader.

-Extended data Figures 2, 3: Provide scale bars on all EM images (micrograph, 2D classes)

-Extended data Figure 2: list absolute number of particles in processing tree. This should match the format of Extended data Figure 3.

Reviewer #2 (Remarks to the Author):

The recycling of RNAP after transcription termination is critical for reusing RNAP and more importantly for resolving the conflict between RNA transcription and DNA replication, but the structural basis for most RNAP-recycling events remains elusive. The current manuscript reports one cryo-EM structure of *B. subtilis* TEC (Transcription elongation complex)-like complex and one cryo-EM structure of *B. subtilis* RNAP core enzyme complexed with an RNAP-recycling factor, HeID. The two structures (determined at 3.36 angstrom) together reveal an unexpected interaction between HeID and RNAP. Such interaction mode induces a large conformational change of RNAP for the release of DNA and RNA. This is a significant advance in the bacterial transcription field. The structures are with good quality and the figure illustrations are well prepared. I suggest acceptance of the manuscript.

Some minor comments:

1. I suggest not naming the first structure as the "transcription elongation complex". Although the cryo-EM map shows DNA/RNA contamination in the RNAP active-center cleft, such observed map is probably from averaging signals of DNA/RNA of different sequences and at different transcription states.

2. I recommend shortening the section for describing the *Bs* RNAP-DNA-RNA complex (p4-p8), as most of the interactions have been described in previously reported bacterial elongation complexes.

3. In the Supplemental Information Figure 1, symbols in the figure that are not specified in the legend. Besides, the residue numbers at right and bottom are not informative.

4. The authors claimed that most HeID-RNAP interface residues are not conserved in the text. How would the non-conserved interface explain similar interaction modes in distinct bacteria?

5. In Supplemental Information Table 3, at current resolution (3.36 angstrom), it is difficult to precisely model side chains of residues, therefore, the information of atom-atom distance is meaningless. I recommend only retaining information of potential interface residues. Moreover, I recommend labeling the RNAP-HeID interface residues in the Supplemental Information Figure 1 to help visualize the conservation pattern of interface residues.

6. Fig. 2a is not convincing and should be removed. It is not comparable of the activities between the *Bs* RNAP core over-expressed *E. coli* cells and the endogenous *Bs* RNAP-HeID. The activities of *Bs* RNAP core itself could be different if prepared in different ways.

7. Previous reports suggest that *Bs* RNAP-delta subunit and HeID facilitate RNAP recycling in a synergistic manner. I am curious why the authors obtained RNAP-HeID complex in an RNAP-delta subunit-deleted *B. subtilis* cells. It would be more informative having the RNAP-delta subunit in the

complex. One or two sentences describing the rationale would be nice.

8. Line 862, "shows an and expanded view of", remove "and"

9. Line 874, the panel letter "d" should be "c"

**Newing et al., Molecular Basis for RNA Polymerase-Dependent
Transcription Complex Recycling By The Helicase-like Motor Protein
Held**

Responses to Referees Comments

REVIEWER COMMENTS

Responses in red

Reviewer #1 (Remarks to the Author):

Newing et al., describe the first cryo-EM structures of *B. subtilis* RNAP in both EC form and bound to the reactivation factor Held. The structures reveal novel aspects of Firmicutes transcription regulation and provide a mechanism for DNA release by Held. The study is of general interest to the transcription community, however, a number of biochemical experiments are missing and are needed to confirm the structural results. Overall, the figures and movies are well rendered. The abstract and discussion could be improved by closing with an overall summary statement, and some phrasing in the paper can be improved. **Changes have been made at the recommended locations and throughout the manuscripts (see marked document).** The reviewer recommends the manuscript for publication in Nature Communications if the authors can address the specific comments below.

Specific comments:

-Figures 1 and 2: the authors prepared the RNAP•Held complex from *B. subtilis* for the experiments in Figure 2. The EC complex in figure 1 was purified by overexpression in *E. coli*. Given that the observed stimulation is far less for the proteins expressed in *E. coli*, the authors should comment on reasons for this observation (perhaps the presence of the epsilon subunit?)

Transcription assays with recombinant RNAP, Held, ATPase-deficient Held (K239A), and natively purified RNAP-Held complexes have been combined into Figure 1a. The assays were repeated *de novo* to construct this new figure panel that also addresses comments below on including a Held only control. In these assays the level of Held-dependent transcription stimulation (assembled recombinant complex vs complex purified from *B. subtilis*) was similar (~2 x). Raw data for the assays is provided in the raw data section.

Additionally, the chromatogram provided in Extended Data Figure 1 indicates that the RNAP purified from *B. subtilis* is not contaminated with nucleic acids, whereas the overexpressed version clearly has nucleic acid contamination. The authors should provide a chromatogram of the *E. coli* expressed protein in addition to that provided for the protein purified from *B. subtilis*.

The EC and HelD complex were not purified in identical ways, and no heparin sepharose chromatography step (shown for the HelD complex in Supplementary Information Fig. 1) was performed with the EC (detailed in the methods). No $A_{280/260}$ chromatographic data is available for the preparation of EC used in this work. We have now analysed the EC and HelD complex using a Qubit fluorometer (data below). The level of nucleic acid determined using the Qubit (HS DNA assay) was low in the EC preparation ($\sim 1/300^{\text{th}}$ the concentration of protein), and probably represents an under-estimation of the actual amount due to the small size of duplex in the complex (10 bp DNA, 8 bp DNA-RNA hybrid) limiting the amount of signal generated through intercalation of the dye. The level of nucleic acid was >15 fold higher in the EC preparation than the HelD complex. We regard the Qubit data a reasonable demonstration of the presence of nucleic acid, albeit with the caveat, it is likely an under-estimation. We consider this likely as examination of EM data shows 3D classes of the EC are well populated with nucleic acids whereas orphan density tentatively attributed to nucleic acid in the HelD-complex was only visible at low thresholds.

Sample	DNA (ng/mL)	Protein (ng/mL)	Sample	DNA:Protein (ng/ng)
Core1	9080	3000000	Core	0.00332
Core2	11300	3840000	HelD-Complex	0.00022
Core3	11060	2640000		
Core Ave	10480	3160000	Core:HelD-Complex	15.4
St Dev	1218	615792		
HelD-Complex1	216	1350000		
HelD-Complex2	326	1460000		
HelD-Complex3	316	1180000		
HelD-Complex Ave	286	1330000		
St Dev	61	141067		

This data is presented in Supplementary Information Table 1, which is now referenced in the main text in L104 and L530.

-Figure 1a/2a Provide a control with only HelD and no RNAP. This is important because malachite green can be used to detect free Pi in solution at low pH. The reviewer is aware that described assays were performed at physiological pH, but it is still critical to show that the observed effects are not due to HelD ATP hydrolysis activity.

We have performed these controls with recombinant wild-type and ATPase-deficient HelD (K239A alteration of ATP binding site), and show the malachite green signal is only due to the transcriptional activity of RNAP. We have also performed ATPase assays with the wild-type and mutant HelD proteins to show the loss of ATPase activity in the mutant. This data is now presented in a modified Figure 1a.

-Figure 1,2,4 : Label upstream and downstream on RNAP. **Done.**

-Methods- Authors refer to “gold standard FSC”. They should state that this has a value of 0.143 and cite Rosenthal and Henderson, JMB 2003. **Done.**

-Lines 227-229 and 311-313: The authors state that the HelD arm interactions with RNAP are mostly based on mechanical force. Given the stability of the complex in high salt buffers, there must be some functional residues or patches. It would greatly help if the authors mutated some of the surfaces, particularly salt bridges or hydrophobic patches, to show which residues are important for the observed association. Proteins do not bind each other randomly. It would also help if specific, relevant amino acid contacts are included in the text. The resolution of the structure should be sufficient to describe these interactions, and the authors have a compiled table that already lists some residues that may be important for the observed interaction.

We appreciate that this is an important study to undertake, but it is clear that it would require a substantial amount of work to ascertain which residues of the protein to target, as even loss of one of the major interaction domains does not prevent stable RNAP interaction. We now state in the manuscript (L226-228), “Previous studies showed that HelD in which the SCA (aa 1-203) had been deleted was still capable of binding RNAP, hydrolysing ATP, and binding DNA, but not transcription recycling Koval *et al.*, 2019”. In addition, the large number of hydrophobic interactions observed will be salt-resistant and may account for the stability of this complex in high salt buffers. Thus, alteration of one, or even multiple residues, is unlikely to prevent HelD binding, or even transcription recycling activity. There is a significant risk that no meaningful results could be obtained within the timeframe necessary for revision of this

manuscript. While this information is desirable, it should be performed as part of a separate project due to the uncertainty of obtaining rapid and meaningful results, and the results obtained from such a study would not change the conclusions drawn from this current work.

-Biochemical assays to show that HelD ATPase activity is necessary for HelD recycling activity. The authors should mutate the HelD ATPase and see if the recycling activity is reduced or association with RNAP is affected. It appears that this may have been done in the accompanying Pei and Kouba manuscripts, but it would help if this data was included in this manuscript.

This work has been performed and addressed in the comments above. As shown in revised Fig 1a, loss of ATPase activity leads to loss of transcription recycling activity in HelD. This is now addressed in the text L236-237.

-Figure 5a: It assumed that this figure is an overlay of the EC structure with the HelD structure. The authors should state this in the figure legend. **Done.**

Additionally, it would help to have a side by side comparison of a normal EC active site versus HelD bound.

This has been done and is presented in revised Figure 5 with panels a and b showing the active site regions in the EC and HelD complex, respectively.

The distortion in the bridge-helix is quite profound.

-Figure 5a: It would help to have an Extended Data figure with a comparison of the HelD positioning relative to GreB/DskA and TFIIS. All use acidic residues to reactivate transcription.

Done. This is now shown in Supplementary Information Fig. 10, and referenced in the text L274-279.

Minor concerns:

-several times the phrase (e.g. lines 23 and 61) “cryo-electron microscopy and single-particle analysis” is used. Generally, “single particle cryo-electron microscopy” is used in the field. **The recommended text change has been made. L23, L60**

-Descriptors like “remarkable” (line 59-60) are a bit over the top for the introduction. **This text has been modified as recommended.** It would be nicer to have a description of the HelD structure rather than using a platitude to describe the protein. **We are unsure what is meant here as we do describe the structure of HelD L63-68.** It would additionally help in the introduction to already indicate that HelD sits on the downstream side of RNAP. **In the**

original submission we do state that HelD is located on the downstream side of RNAP (L53-54).

-lines 42-44 somewhat redundant with the use of recycling. The last phrase “indicating it....recycling RNAP” can be left out. **This text has been modified as recommended.**

-line 112- show a supplementary figure with overlays of B. subtilis RNAP, E. coli RNAP, and M. smegmatis, etc. ECs. This would make the authors comparison more clear to the reader.

Supplementary Information Figure 4 has been modified to include images of complexes from E. coli and M. smegmatis. Presentation of overlaid structures was unsatisfactory so similar orientations of the representative RNAPs are shown with lineage-specific inserts shown in red.

-Extended data Figures 2, 3: Provide scale bars on all EM images (micrograph, 2D classes) **Done**

-Extended data Figure 2: list absolute number of particles in processing tree. This should match the format of Extended data Figure 3. **Done. These are now Supplementary Information Figures 2 and 3.**

Reviewer #2 (Remarks to the Author):

The recycling of RNAP after transcription termination is critical for reusing RNAP and more importantly for resolving the conflict between RNA transcription and DNA replication, but the structural basis for most RNAP-recycling events remains elusive. The current manuscript reports one cryo-EM structure of B. subtilis TEC (Transcription elongation complex)-like complex and one cryo-EM structure of B. subtilis RNAP core enzyme complexed with an RNAP-recycling factor, HelD. The two structures (determined at 3.36 angstrom) together reveal an unexpected interaction between HelD and RNAP. Such interaction mode induces a large conformational change of RNAP for the release of DNA and RNA. This is a significant advance in the bacterial transcription field. The structures are with good quality and the figure illustrations are well prepared. I suggest acceptance of the manuscript.

Some minor comments:

1. I suggest not naming the first structure as the “transcription elongation complex”. Although the cryo-EM map shows DNA/RNA contamination in the

RNAP active-center cleft, such observed map is probably from averaging signals of DNA/RNA of different sequences and at different transcription states.

We do believe that the structure represents a transcription elongation complex and that it would not be appropriate to change the naming of the structure. The active site region including the BH has local resolution of $\sim 3.2\text{\AA}$ and all the structural elements within the active site region are of similar high resolution (with the exception of the flexible TL). Ribose phosphates are well resolved and defined in the nucleic acids, but the bases are not specific due to how the complex was purified. Thus, there is no ambiguity over the conformational state of the EC. The register of the nucleic acids, along with the conformation of BH and TL are all fully consistent with the interpretation of the data as an EC in the post-translocation conformation.

2. I recommend shortening the section for describing the Bs RNAP-DNA-RNA complex (p4-p8), as most of the interactions have been described in previously reported bacterial elongation complexes.

This section has not been shortened as much of the text is involved in the description of novel features of *B. subtilis* RNAP (*e.g.* the β In5 domain and ϵ subunit). This is an important reference structure for medically important bacteria (*e.g.* *Staphylococci*, *Streptococci*, *Enterococci*, *Clostridia*), against which anti-transcription drugs are used (*e.g.* fidaxomicin against *C. difficile*), and so we feel a detailed description of the data is warranted.

3. In the Supplemental Information Figure 1, symbols in the figure that are not specified in the legend. Besides, the residue numbers at right and bottom are not informative.

The figure legend (now Supplementary Information Figure 7) has been modified. The symbols (histogram?) below the alignment was mentioned at the end of the original legend, but this has been moved towards the top of the revised legend. The meaning of the numbers to the left, right and bottom of the alignment has also been clarified. This has also been included in the legend to Supplementary Information Figure 8 with the alignment of *L. plantarum* and *B. subtilis* HelD sequences.

4. The authors claimed that most HelD-RNAP interface residues are not conserved in the text. How would the non-conserved interface explain similar interaction modes in distinct bacteria?

We have partially addressed this issue in response to comments from Referee 1. From what we currently know, based on two structures and sequence

alignment, there do not appear to be many residues involved in specific interaction of *B. subtilis* RNAP and HelD that are conserved. The structure of the SCA and CA are likely to be important in facilitating HelD binding to RNAP permitting species-specific interaction.

5. In Supplemental Information Table 3, at current resolution (3.36 angstrom), it is difficult to precisely model side chains of residues, therefore, the information of atom-atom distance is meaningless. I recommend only retaining information of potential interface residues. Done (now Supplementary Information Table 5).

Moreover, I recommend labeling the RNAP-HelD interface residues in the Supplemental Information Figure 1 to help visualize the conservation pattern of interface residues.

This is now Supplementary Information Figure 7. The residues involved in formation of salt-bridges and hydrogen bonds have been indicated using black arrowheads and the figure legend adjusted accordingly.

6. Fig. 2a is not convincing and should be removed. It is not comparable of the activities between the Bs RNAP core over-expressed *E. coli* cells and the endogenous Bs RNAP-HelD. The activities of Bs RNAP core itself could be different if prepared in different ways.

Fig 2a has been removed, and the figure 2 modified. The difference between activities of native and recombinant complexes has been addressed in comments to Referee 1.

7. Previous reports suggest that Bs RNAP-delta subunit and HelD facilitate RNAP recycling in a synergistic manner. I am curious why the authors obtained RNAP-HelD complex in an RNAP-delta subunit-deleted *B. subtilis* cells. It would be more informative having the RNAP-delta subunit in the complex. One or two sentences describing the rationale would be nice.

We do state on L190-192 that the delta subunit is absent in many organisms that contain HelD, and this includes the *Clostridia/Clostridiodes* that are closely related to *B. subtilis*. In addition, HelD is able to catalyse transcription complex recycling in the absence of delta (reference #12; and Fig 1a, this work). Delta is more abundant than HelD within the cell, and is also implicated in multiple other roles in modulating transcription activity. Therefore, to understand the mechanism of HelD-catalysed recycling of transcription complexes, and ensure we were working with a homogeneous system, we felt it was valid to isolate complexes free of delta.

An accompanying paper to ours from the laboratory of Prof Markus Wahl does include structures that include delta and is referenced accordingly in our manuscript.

8. Line 862, “shows an and expanded view of”, remove “and”. Done (now L913)

9. Line 874, the panel letter “d” should be “c”. Done (now L923).

REVIEWERS' COMMENTS

Reviewer #1 (Remarks to the Author):

Newey et al have addressed all reviewer concerns. The manuscript is suitable for publication.